# The structure of plastocyanin tunes the midpoint potential by restricting axial ligation of the reduced copper ion

Claire C. Mammoser [1], Brynn E. LeMasters [1,2], Sydney G. Edwards[1], Emma M. McRae [1], M. Hunter Mullins[1,3], Yiqi Wang[1,4], Nicholas M. Garcia [1,5], Katherine A. Edmonds[1], David P. Giedroc[1] & Megan C. Thielges [1]✉

Blue copper proteins are models for illustrating how proteins tune metal properties. Nevertheless, the mechanisms by which the protein controls the metal site remain to be fully elucidated. A hindrance is that the closed shell Cu(I) site is inaccessible to most spectroscopic analyses. Carbon deuterium (C-D) bonds used as vibrational probes afford non-perturbative, selective characterization of the key cysteine and methionine copper ligands in both redox states. The structural integrity of *Nostoc* plastocyanin was perturbed by disrupting potential hydrogen bonds between loops of the cupredoxin fold via mutagenesis (S9A, N33A, N34A), variably raising the midpoint potential. The C-D vibrations show little change to suggest substantial alteration to the Cu(II) coordination in the oxidized state or in the Cu(I) interaction with the cysteine ligand. They rather indicate, along with visible and NMR spectroscopy, that the methionine ligand distinctly interacts more strongly with the Cu(I) ion, in line with the increases in midpoint potential. Here we show that the protein structure determines the redox properties by restricting the interaction between the methionine ligand and Cu(I) in the reduced state.

[1] Indiana University Department of Chemistry, 800 E. Kirkwood Ave., Bloomington, IN 47405, USA. [2] Present address: Department of Chemistry, University of Wisconsin, Madison, WI 53706, USA. [3] Present address: Indiana University School of Medicine, Indianapolis, IN 46202, USA. [4] Present address: Department of Chemical and Biomolecular Engineering, Johns Hopkins University, Baltimore, MD 21218, USA. [5] Present address: University of Wisconsin School of Medicine and Public Health, Madison, WI 53726, USA. ✉email: thielges@indiana.edu

Blue copper proteins (BCPs) serve as textbook models for illustrating how proteins tune the chemical and physical properties of metal sites[1,2]. In type I BCPs, the copper (Cu) ion typically is coordinated through strong interactions with three residues - two histidines and a cysteine, and by weak interaction with methionine and sometimes a second axial ligand - resulting in a distorted tetrahedral active site. Extensive efforts have previously uncovered much about how the redox properties arise from the characteristic coordination geometry. Nonetheless, the precise role played by the greater protein structure has not been rigorously elaborated. Moreover, a major obstacle to fully elucidating how redox properties are tuned by proteins is their dependence on both the oxidized and reduced states. Experimental characterization of the reduced states of BCPs is limited since electronic-based spectroscopic techniques cannot access the $d^{10}$ closed shell system of Cu(I). Here we perturb the redox potential of the BCP plastocyanin (Pc) by disrupting the structure through mutagenesis. We then use selectively introduced carbon-deuterium (C-D) bonds as vibrational probes of metal-ligand bonding to perform a comprehensive analysis of any associated changes in the interaction between both the critical cysteine and methionine ligands and the Cu ion in both redox states.

The primary coordination sphere (Fig. 1a), particularly the Cu ion's strong interaction with the cysteine and weak interaction with the axial methionine ligand, predominantly underlies the physical and redox properties of BCPs[3–6]. The strong interaction between the cysteine thiolate and Cu(II) ion results in the characteristic visible absorbance at ~600 nm underlying the signature blue color of oxidized BCPs. The coordination geometry is intermediate between ideal tetrahedral and square planar geometries preferred, respectively, by Cu(I) and Cu(II) small molecule complexes. The bond lengths and angles change only subtly between the two redox states, underlying the low reorganization energies[7,8]. The ligand coordination also provides a means of tuning the redox potential. In general, the midpoint redox potential ($E_m$) is elevated (~360 mV vs. NHE for plastocyanin) compared to the standard aqueous redox potential of copper (Cu(I/II)) (161 mV vs. NHE[9]) but can range by >500 mV among the BCP family[6,10,11].

Protein homology analysis and extensive mutational studies of the ligand set forming the primary coordination sphere has established that their identity is the critical determinant of the physical and redox properties of BCPs[5,12,13]. However, the potential influence of the rest of the protein and surrounding solvent, the outer coordination sphere, also is well recognized but highly debated[6,14–17]. A long-established proposal is that the coordination geometry is under strain imposed by the protein in an entatic or rack-induced state[18–20]. According to these classic models, the protein maintains a geometry intermediate to those preferred by the oxidized and reduced metal site[19,20]. A revision to this model is that the distorted geometry of the oxidized state is not imposed but inherently preferred because of the high polarizability of the cysteine and methionine ligands[21,22]. Alternately, some studies argue that the Cu-methionine bond would break in the absence of constraint by the protein[5,23], while others argue that the protein functions to reduce the interaction between the Cu ion and the axial ligand, keeping the bonding weak[4,24]. Another proposal is that the methionine side chain is flexible due to a shallow potential and can adjust between the oxidation states with little energetic penalty[25,26].

BCPs share a conserved cupredoxin fold, composed of eight β-strands folded in a Greek-key topology into two β-sheets that form a β-barrel tertiary structure (Fig. 1b, c)[27–29]. Three of the Cu ligands - Cys89, Met97, and His92 for *Nostoc* Pc - are located on a single loop. Mutations that alter residues that tether this loop are perturbative to the Cu site[6,10,16,17]. For example, mutation of Asn38 in poplar Pc, which interacts with the cysteine ligand, significantly destabilizes the Cu site[14,17]. As intramolecular interactions can relay structural constraints over long distances[30,31], the rest of the cupredoxin structure that does not directly interact with the Cu ligands could also contribute to the Cu site properties but is less well investigated. Treatment with denaturants leads to a new state with higher $E_m$ but no change in the oxidized visible spectrum, arguing by elimination that the reduced state of BCPs is selectively stabilized[24].

To investigate whether and how the greater structure influences the redox properties of BCPs, we introduced mutations intended to remove hydrogen bonding interactions between the loops that connect the β-strands at the northern side of the cupredoxin fold. We identified in the x-ray structure of *Nostoc* Pc (PDB: 2GIM) three residues, Ser9, Asn33, and Asn34, with side chains that participate in hydrogen bonding within or between the loops and mutated them to alanine (Fig. 1b). These residues are indicated to form hydrogen bonds across multiple crystals- and solution-phase structures of Pc, although the exact number and placement of these bonds vary, and Ser9 is not conserved across all higher plants (Fig. S1, Fig. S2)[32–34]. We measured the $E_m$ to be elevated for all mutated Pc.

We then characterized any changes in the Cu-ligand interactions using C-D vibrational labels by incorporating (3,3-$d_2$)-cysteine and (methyl-$d_3$)-methionine at the Cu ligands Cys89 and Met97 ($d_2$Cys89 and $d_3$Met97) (Fig. 1a). C-D vibrations absorb in a transparent window of protein spectra where no native vibrations absorb, hence their absorptions can be detected distinctly from the congestion of complex infrared (IR) spectra of proteins[35,36]. They introduce no perturbation, which is essential

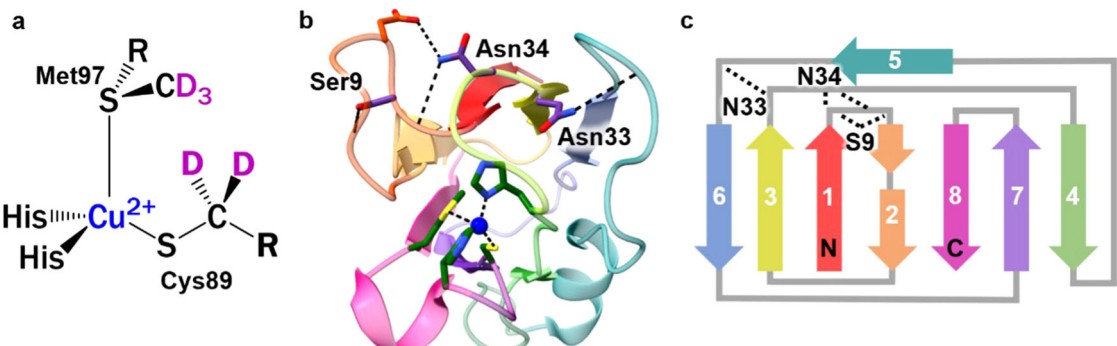

**Fig. 1 Structural features of *Nostoc* plastocyanin (Pc). a** Chemical structure of primary coordination sphere. Deuterium labels are highlighted in magenta. **b** Ribbon model of crystal structure (PDB: 2GIM) with side chains shown for Cu ligands and mutated residues, and hydrogen bonds to loops indicated with dashed lines. **c** Topology map with the hydrogen bonds between β-strands indicated with dashed lines.

for the study of delicate metal sites in proteins. We previously have shown that C-D vibrations at Cys89 and Met97 in Pc are highly sensitive to the interaction of the ligands with the Cu ion[37,38].

The C-D vibrations report no significant change in the Cu-Cys89 bonding in either the redox state or for either ligand in the oxidized state. Distinctly, the C-D vibrations at Met97 in the reduced state indicate stronger interaction with the Cu(I) ion that correlates with the increases in $E_m$. Nuclear magnetic resonance (NMR) spectroscopy of uniformly [15]N-labeled Pc in the reduced state corroborates mutation-induced perturbation to the northern loops of the cupredoxin fold. This study provides experimental evidence that the structure of Pc functions to restrict the axial methionine ligand in a way that reduces its interaction with the reduced Cu(I) ion, resolving a long-standing question about entatic control in BCPs.

## Results

**Characterization of mutated Pc preparations.** Missense mutations coding S9A, N33A, N34A, and N40A were successfully introduced into plasmid pEAP-MAAE for expression of *Nostoc* Pc, as confirmed by DNA sequencing. S9A, N33A, and N34A Pc were successfully expressed in JM15(DE3) *E. coli* as determined by SDS PAGE (Fig. S3) and MALDI-ToF mass spectrometry (Fig. S4). N40A Pc however was expressed as a colorless protein and failed to gain the blue color characteristic of the Cu(II) site of Pc even after extensive dialysis with buffer containing 200 μM CuSO4. N40A Pc could not be purified in sufficient quantity for additional characterization. S9A, N33A, and N34A Pc could be purified, but the preparation only fully gained blue color after extensive dialysis for 2–4 days in buffer containing CuSO4, in comparison to 1–2 days for wild-type (wt) Pc. This could indicate slower oxidation of the mutated proteins due to their elevated $E_m$s (vide infra). Alternately, similar observations were reported for heterologous expression of *Pseudomonas aeruginosa* azurin in *E. coli* and attributed to mis-metalation with Zn(II) and slow metal exchange by Cu(II)[39]. Spectrometric assay of the final preparations using 1-(2-Pyridylazo)-2-naphthol as a Cu(II) chelator was used to assess Cu(II) content (Fig. S5). This assay reports ~0.7 Cu(II) per polypeptide chain for wt Pc. The value less than one is likely due to incomplete denaturation of the holoprotein during the procedure. In comparison, the assay reports ~0.9-1 Cu(II) per polypeptide chain for the mutated Pc. Mass spectrometry showed no peaks corresponding to apoprotein, indicating fully metallated wt and mutated proteins (Fig. S4).

Upon titration of ascorbate into a solution of oxidized protein in potassium ferricyanide, which acts as a redox buffer, the 597 nm absorption disappears, indicating a reduction of the Cu(II) to Cu(I) in Pc (Supplementary Data 1). Nernst plots of the ratio of Pc redox species vs. cell potential show linear variance that was fit to determine the $E_m$s. Compared to wt, all mutated constructs (S9A, N33A, and N34A) show higher $E_m$s (Table 1, Fig. S6), although we note that the error bounds overlap for wt

and S9A. The greatest change in $E_m$ is a 78 mV increase for N33A Pc.

*UV/visible spectroscopy.* UV/visible absorption spectra of the wt and mutated Pc in the oxidized state show a strong absorbance at 597 nm (16,750 cm$^{-1}$) assigned to the Cys89/Cu(II) charge transfer band characteristic of oxidized Pc (Fig. 2a)[8]. The indistinguishable visible spectra among the wt and mutated proteins support that the electronic structure of the oxidized Cu(II) site, particularly the covalent bonding between the Cu(II) and Cys89 ligand, is not significantly perturbed by the mutations.

*Amide I IR spectroscopy.* The IR spectra of wt and all mutated Pc in D$_2$O in the oxidized or reduced states show an absorption envelope around 1600–1700 cm$^{-1}$ consistent with previous reports of amide I spectra of BCPs (Fig. 2b, c)[40,41]. No significant differences were observed among the spectra of the wt and mutated Pc, indicating that the mutations do not substantially disrupt the secondary structure. Overlay of the second derivative spectra does indicate a possible shift in frequencies of minor amide I band for the oxidized N33A Pc, suggesting some perturbation. The second derivative spectra of wt and each mutated Pc are nearly superimposable for the reduced state.

*NMR spectroscopy.* 2D NMR spectroscopy was used to assess how the mutations perturb the structure of Pc in the reduced state. [1]H,[15]N heteronuclear single quantum coherence (HSQC) spectra of uniformly [15]N-labeled wt and N33A Pc (Fig. 3a, Fig. S7, Supplementary Data 2) show that the chemical shifts of resonances for most residues are unperturbed, indicating that the overall cupredoxin structure is unaltered. However, a number of residues located in loops show significant chemical shift perturbations (Fig. 3b). These include mutated residue Asn33, as well as the His39 and Met97 ligands, and locations spanning the northern face of the cupredoxin fold (Fig. 3c). NMR spectroscopy, therefore, supports that mutation of Asn33 disrupts inter-loop hydrogen bonding, destabilizing the native structure of the northern face of the cupredoxin fold as intended, while maintaining the beta barrel core. Details of this analysis are provided in Supplementary Methods.

## IR spectroscopy of C-D vibrational probes

The FT IR spectra of d$_3$Met97/d$_2$Cys89 for freshly prepared oxidized wt Pc and chemically reduced protein show absorptions in the transparent frequency window, as previously characterized[37,38]. Our analysis focused on the absorptions of the d$_2$Cys89 CD$_2$ asymmetric stretch and d$_3$Met97 CD$_3$ symmetric stretch modes due to their greater intensity (Supplementary Data 3). Although the absorptions of C-D vibrations at d$_2$Cys89 are weak (~2.5 cm$^{-1}$ M$^{-1}$) (Fig. S8), they are extremely sensitive to interaction with the Cu ion. The frequency of the asymmetric stretching mode of the C-D vibration of d$_2$Cys89 decreases from 2235 cm$^{-1}$ for the oxidized protein to 2209 cm$^{-1}$ for the reduced protein, a ~26 cm$^{-1}$

**Table 1 Center frequencies from Gaussian fits to C-D spectra and corresponding $E_m$ values$^a$.**

| | $\nu_{as}$ [d$_2$Cys89] (cm$^{-1}$) | | $\nu_s$ [d$_3$Met97] (cm$^{-1}$) | | $E_m$ (mV, vs. NHE) |
|---|---|---|---|---|---|
| | Cu(II) | Cu(I) | Cu(II) | Cu(I) | |
| wt | 2235.1 (0.2) | 2209.4 (0.3) | 2126.30 (0.02) | 2123.31 (0.05) | 362 (10) |
| S9A | 2234.9 (0.4) | 2209.7 (0.1) | 2126.30 (0.01) | 2123.8 (0.1) | 371 (7) |
| N34A | 2235.2 (0.3) | 2209.81 (0.08) | 2126.50 (0.03) | 2123.92 (0.08) | 389 (9) |
| N33A | 2234.7 (0.3) | 2208.8 (0.4) | 2126.23 (0.03) | 2124.3 (0.1) | 440 (6) |

$^a$Standard deviation from triplicate data given in parentheses.

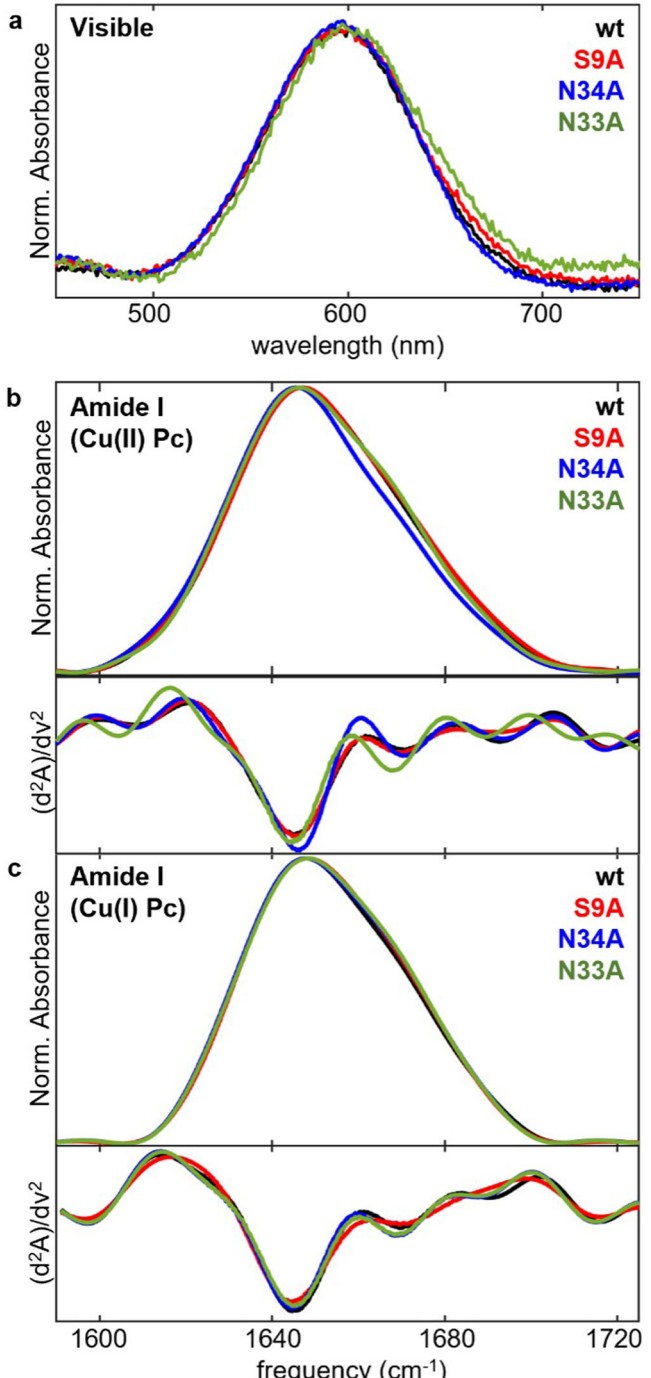

**Fig. 2 Spectroscopic analysis of wt and mutated plastocyanin. a** Visible absorption characteristic of oxidized BCPs. **b** Amide I IR absorption of oxidized proteins in $D_2O$. **c** Amide I IR absorption of reduced proteins in $D_2O$. Second derivative spectra shown in lower panels of (**b**) and (**c**).

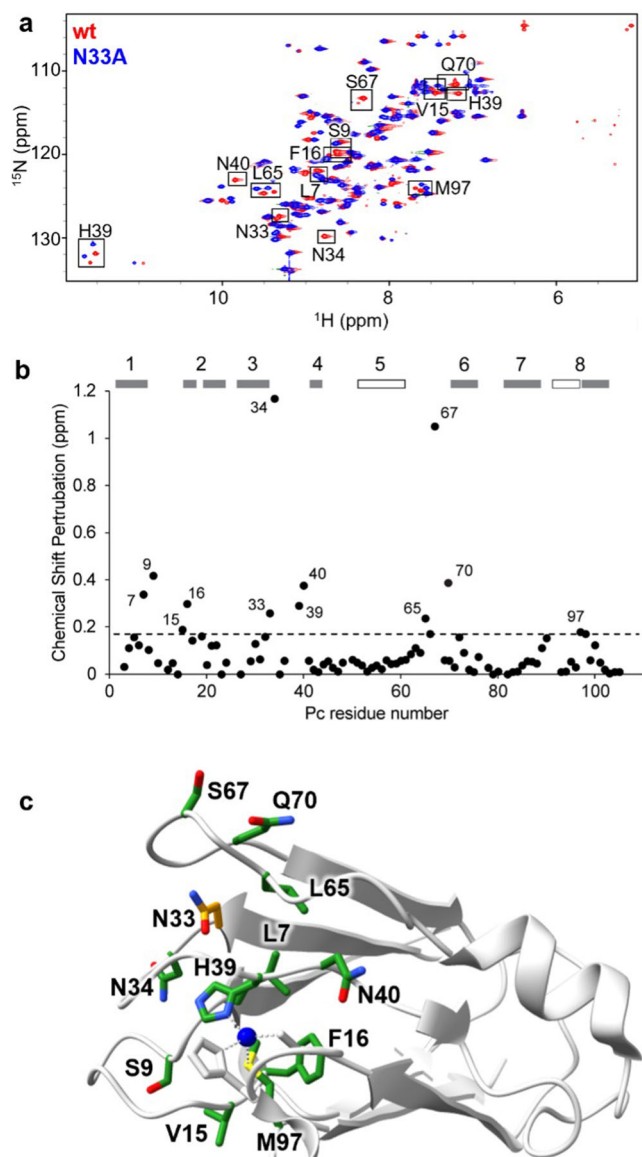

**Fig. 3 NMR spectroscopic analysis of wt and N33A plastocyanin.**
**a** Overlaid $^1H$-$^{15}N$ HSQC spectra of wt (red) and N33A (blue) Pc. Boxed resonances in the spectrum of wt Pc have an estimated chemical shift perturbation of more than 0.17 ppm upon mutation of N33 to A.
**b** Estimated chemical shift perturbations between wt and N33A, with secondary structure indicated above as gray bars for beta strands and open bars for helices. The threshold of 0.17 ppm is shown as a dashed line, and residues with chemical shift perturbation above that threshold are labeled.
**c** Side chains of residues corresponding to boxed resonances in (**a**) are shown in green on the wt crystal structure (PDB: 2GIM). N33 is shown in orange.

difference between the redox states. Despite this sensitivity, the spectra of $d_2$Cys89 for all the mutated constructs of Pc were essentially indistinguishable from wt for both oxidized and reduced states, shifting by at most a couple tenths of a cm$^{-1}$ (Table 1, Fig. 4a, Table S1).

The absorption of the C-D symmetric stretching mode of $d_3$Met97 is more intense (~30 cm$^{-1}$ M$^{-1}$) than those of $d_2$Cys89 but less sensitive to the interaction with the Cu ion. The C-D frequency for $d_3$Met97 decreases from 2126.3 cm$^{-1}$ for the oxidized protein to 2123.3 cm$^{-1}$ for the reduced protein, a

3 cm$^{-1}$ difference between the redox states (Fig. 4b). However, it shows little change (at most 0.2 cm$^{-1}$) due to any of the mutations for the oxidized protein (Fig. 4c). Only a slight variance in linewidths is observed among the mutated proteins (Table S2). Generally, the C-D probes report little impact on the interaction of $d_3$Met97 with Cu(II).

Distinctly, the C-D vibration of $d_3$Met97 is higher in frequency for the mutated Pc when in the reduced state (Fig. 4d). For N33A Pc, the frequencies are higher than wt by 1 cm$^{-1}$. While this change appears modest, the magnitude is well outside error and a

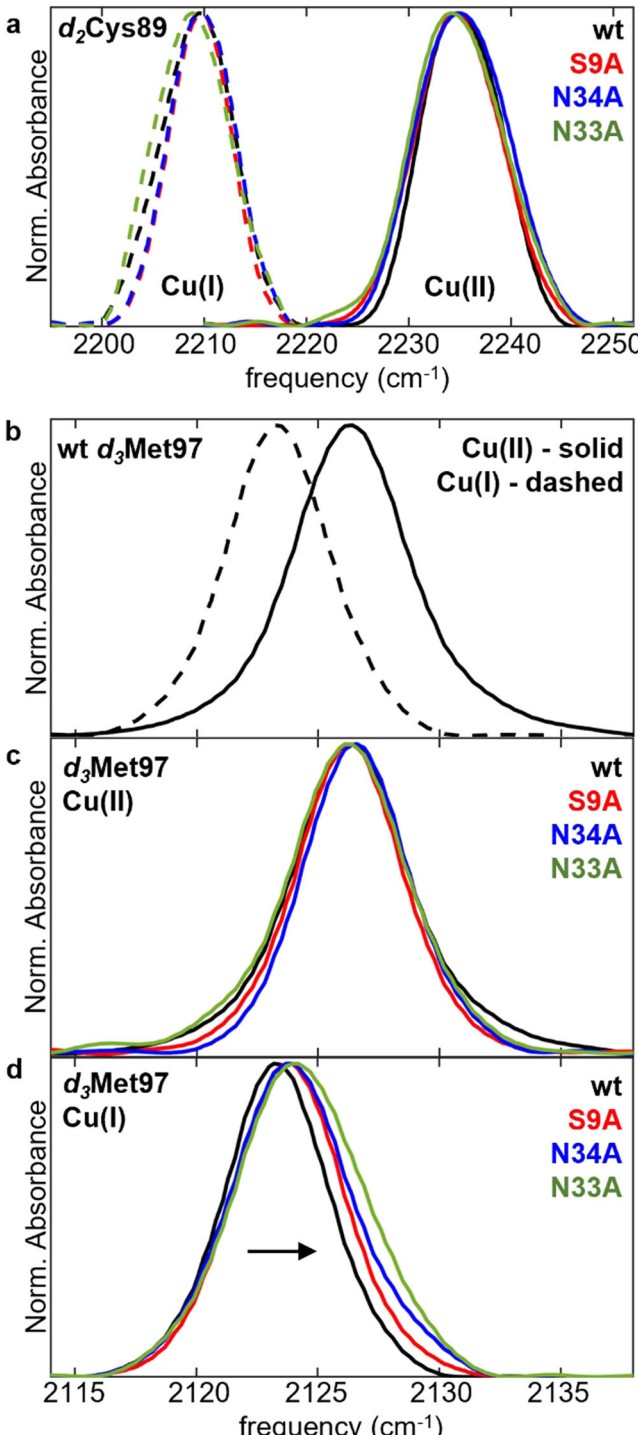

**Fig. 4 IR spectral analysis of C-D vibrational probes in plastocyanin.**
**a** Overlays of IR spectra of the C-D asymmetric stretch of $d_2$Cys89 of oxidized (solid) and reduced (dashed) Pc. **b**–**d** Overlays of IR spectra of the C-D symmetric stretch of $d_3$Met97 for (**b**) wt oxidized (solid) and reduced (dashed) Pc; and wt and mutated (**c**) oxidized and (**d**) reduced Pc. Arrow in (**d**) demarks shift in the band associated with mutations.

third of the difference between the redox states. The frequencies for S9A and N34A Cu(I) Pc are intermediate between wt and N33A Pc. Thus, the C-D frequency of $d_3$Met97 for the mutated Pc in the reduced state generally correlates with the $E_m$s (Table 1, Fig. 5a). The linewidths of the C-D absorptions also correspondingly increase (Fig. S9, Table S2). Alternative fitting models are

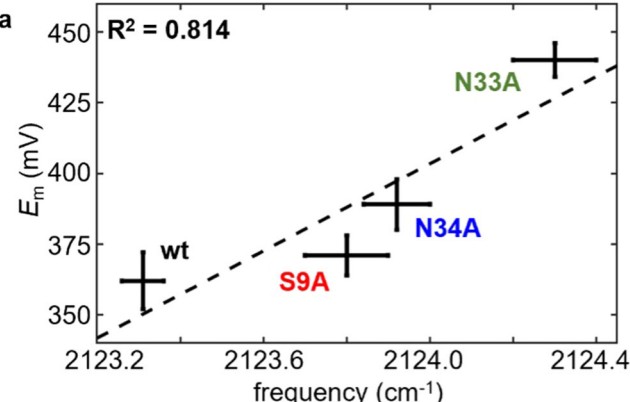

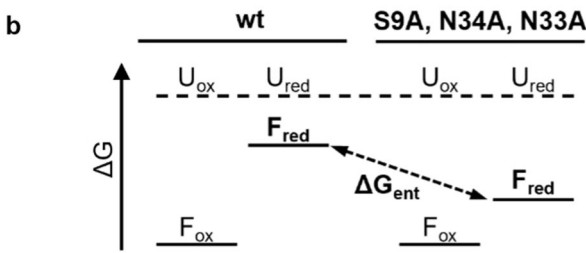

**Fig. 5 Relationship between Cu(I) coordination and $E_m$. a** Plot of $E_m$ and corresponding frequency of C-D symmetric stretch vibration of $d_3$Met97 for reduced wt and mutated Pc. Error bars reflect standard deviation from triplicate measurements. **b** Model of changes in free energy of metal ion coordination (ΔG) due to mutations consistent with experimental data. $U_{ox/red}$, $F_{ox}$, and $F_{red}$ refer, respectively, to the Cu site of the unfolded oxidized/reduced, folded oxidized, and folded reduced protein. $\Delta G_{ent}$ is the change in free energy associated with constraining the Cu(I) site that is disrupted by the mutations.

considered in Supplementary Methods, Fig. S10, and Table S3, but do not alter the correlation among the mutation-induced spectral changes and $E_m$.

**Additional results indicating that destabilization of Pc structure stabilizes a reduced metal site.** We also noted in the course of this study that oxidized samples of the mutated proteins that had been flash-frozen and subsequently thawed show a significant absorption band at 2210 cm$^{-1}$ that is indistinguishable from the band observed for $d_2$Cys89 of reduced wt Pc (Fig. S11). The integrated band areas indicate a population of reduced protein of ~35% for S9A and N34A Pc and ~70% for N33A Pc. The mutations thus make Pc susceptible to forming a new conformational state with higher $E_m$. We have been unable to reliably characterize the C-D absorptions of $d_2$Cys89 of the apoprotein or unfolded protein, presumably due to the low extinction coefficient and increased breadth of the absorption when not ligated to a metal ion. Spectra of $d_4$-cystine in 500 mM HCl show a single absorbance centered at ~2244 cm$^{-1}$ (Fig. S8). We thus attribute this freezing-induced population to a distorted state rather than apoprotein or unfolded protein. A band at the frequency of reduced protein (2210 cm$^{-1}$) also appears for wt Pc in the presence of urea (Fig. S12). Furthermore, this band appears over time (~40 days) when wt or mutant proteins are incubated at 4 °C (Fig. S13, Table S4). The reduction of the protein appears reversible, as the addition of oxidant potassium ferricyanide and redox mediators yields an increase in the 600 nm visible

absorbance (Fig. S14). Further description of these experiments is provided in Supplementary Methods.

## Discussion

Here we investigate the role of the cupredoxin structure in determining the properties of Cu sites in BCPs. We introduced mutations (S9A, N33A, N34A) designed to disrupt hydrogen bonding interactions that tether the loop structures that form the northern face of the cupredoxin fold of *Nostoc* Pc (Fig. 1b)[33] and found all to varying extents increase the $E_m$. S9A is associated with a small increase in the $E_m$ from wt of 9 mV, although the change is within error bounds. Consistently, S9A is intended to remove a hydrogen bond within a turn connecting contiguous β-strands 1 and 2 (Fig. 1c) and lead to localized disruption. Ser9 is also the least conserved among the mutated residues[42]. Larger increases in $E_m$ of 27 mV and 78 mV result from N33A and N34A, respectively. These mutations are intended to disrupt hydrogen bonds between discontinuous parts of the structure. Based on the crystal structure[33], Asn34 tethers the loops between strands 1 and 2 and strands 3 and 4. Asn33 tethers the loops between strands 3 and 4 and strands 5 and 6 and is located near Cu ligand His39.

IR analysis of the C-D probes at the Cys89 and Met97 ligands indicates that altered Cu(II) coordination of oxidized Pc does not underlie the increased $E_m$. In prior studies, we found the frequency of C-D vibrations of ligand $d_2$Cys89 particularly sensitive to interaction with the metal ion. The frequency shifts by 26 cm$^{-1}$ between the redox states and varies by as much as 45 cm$^{-1}$ from the individual, reduced Pc to the oxidized Pc in complex with the redox partner cytochrome $f$[38]. C-D probes at Met97 are also modestly sensitive to changes at the metal site, differing by 3 cm$^{-1}$ between redox states and up to 5.5 cm$^{-1}$ for metal-substituted Pc[37]. We previously demonstrated that C-D vibrations at the Cys89 and Met97 ligands are impacted by redox state, partner binding, metal substitution, demetallation, and unfolding[37,38]. Nonetheless, the vibrations display little change among oxidized wt and S9A, N33A, or N34A Pc, arguing that the Cu(II) coordination by the Cys89 and Met97 ligands is not significantly perturbed. This conclusion is corroborated by the unvarying visible charge transfer band appearing at 597 nm (Fig. 2a) that arises from the strong covalent Cu(II)-Cys89 interaction.

Any perturbations by the mutations to the oxidized protein are subtle. The amide I vibrational absorption (Fig. 2b, c) indicates maintenance of the β-sheet core. The possible shifts in the amide I frequencies discerned in the second derivative spectra for the oxidized state of N33A relative to wt and other mutated Pc suggest some minor structural impact. In addition, slight variation in side chain flexibility of $d_3$Met97 is supported by the C-D absorption linewidths that reflect the heterogeneity sensed by the probe. Generally, however, the data argue that the mutations do not substantially impact the oxidized state.

Distinctly, the C-D vibrations of $d_3$Met97 reveal that the mutations alter the interaction between the axial ligand and the Cu(I) ion of reduced Pc. No corresponding changes are found for $d_2$Cys89 of the reduced protein. Based on our prior analysis[37,38], the higher C-D frequencies found for the mutated Pc are indicative of stronger interaction between the metal ion and Met97. This relationship is illustrated, for example, by the higher C-D frequency of $d_3$Met97 when interacting with the more highly charged Cu(II) versus the Cu(I) ion (Table 1). Stabilization of the reduced metal site through the stronger Cu(I)-Met97 interaction is predicted to raise $E_m$. This prediction is in line with the experimental data; the increases in C-D frequency of $d_3$Met97 among the mutated Pc generally correlate with the increases in

$E_m$ (Table 1, Fig. 5a). Corresponding with the higher C-D frequencies and an increase in $E_m$ is an increase in the absorption linewidths (Fig. S9). The broader linewidths reflect greater heterogeneity experienced by the Met97 side chain, a change consistent with increased flexibility due to the loss of constraint by the protein.

The free energy diagram in Fig. 5b summarizes the resulting model for how the protein scaffold controls the stability of the Cu site and thereby $E_m$ in Pc. This is essentially the model proposed for the increased $E_m$ due to denaturants but that could not be confirmed experimentally[24,43,44]. The $E_m$ and associated free energy for the reduction of the metal site for the unfolded proteins can be assumed to be largely unaffected by the mutations to the native structure; the unfolded proteins serve as arbitrary references in Fig. 5b. The oxidized Cu(II) site of the folded protein is not significantly affected by the mutations as demonstrated by the visible and IR spectroscopic data, and is more stable than the reduced Cu(I) site[45]. The mutations lead to higher $E_m$, corresponding to more favorable free energy for forming the reduced state. Since the free energy of the other states can be considered fixed, this implies stabilization of the reduced Cu(I) site in the mutated proteins. The IR data show that this stabilization is mediated by stronger interaction with the axial ligand Met97. Conversely, the results imply that the structure of the wt protein constrains the Met97 side chain in a way that weakens the Cu(I)-Met97 interaction of the reduced state. The largest change in $E_m$ of 78 mV for N33A indicates the free energy of constraining the Cu site is at least 7.5 kJ/mol.

NMR spectroscopy confirms that the N33A substitution only locally alters the structure of reduced Pc. Overlay of $^1$H$^{15}$N HSQC spectra of $^{15}$N-enriched wt and N33A Pc indicates that the mutation leaves the beta barrel core intact while disrupting the loops at the northern face. Residues that show significant chemical shift perturbation between wt and N33A include Asn33 itself and residues of the loop between strands 5 and 6 (Leu65, Ser67, Gln70); these perturbations are fully consistent with our objective of disrupting only the inter-loop hydrogen bonds present in the crystal structure. In addition, the mutation-induced chemical shift perturbations span the loops of the northern face of the cupredoxin fold. The disruption from mutation of Asn33 propagates through Cu ligand His39 and secondary shell residue Asn40 across the Cu site to Met97 and surrounding hydrophobic residues Val15 and Phe16 (Fig. 3c). The change to a single amino acid is relayed across the protein structure to affect the primary coordination and properties of the metal center.

Although stronger axial ligation mostly explains the variation in $E_m$ among the mutated Pc, another factor to consider is electrostatics[46–48]. Structural disruption could alter the solvent exposure of the metal site. An effective increase in dielectric constant would be stabilizing for the more highly charged species, predicting a decrease in $E_m$, as opposed to the overall increase observed experimentally. Greater hydrophobicity in the local environment would increase $E_m$, as has been found by axial ligand replacement[49,50], a possibility in a misfolded state. Greater hydrophobicity surrounding the Met97 side chain would contribute to a downshift in the C-D frequency of $d_3$Met97[51], opposite the overall trend among the mutated Pc. The mutations themselves also change the electrostatic environment, as each substitutes a polar side chain with alanine. These effects are expected to be relatively minor, however, as more extreme charge removal or reversal mutations introduced in previous studies in *Nostoc* Pc[52,53] and other BCPs[54–58] induce smaller changes in $E_m$ than observed here. Notably, our prior analysis of the C-D probes in Pc indicates they are primarily sensitive to the ionic interaction with the metal ion[37,38]. Nonetheless, electrostatic factors may

contribute to the deviation from the observed relationship between $E_m$ and C-D frequency of $d_3$Met97 (Fig. 5a).

The strengthening of the Cu(I)-Met97 interaction of the reduced protein revealed by the spectroscopic data well explains the higher $E_m$s. Extensive mutation of the axial ligand in Pc, as well as other BCPs, with natural and unnatural amino acids, have deconvoluted how the donor strength and other properties of the axial ligand tune the metal site[49,50,59,60]. Increased donor strength generally results in decreased $E_m$ because the more highly charged oxidized state is selectively stabilized. This relationship is observed also in purple CuA centers[61,62], although the impact of axial ligand mutation is less than in BCPs. In contrast, our study of Pc finds that neither the interaction between Met97 with the oxidized Cu(II) nor between Cys89 and the Cu ion in either redox state are affected by the loop mutations. These interactions are stronger than Cu(I)-Met97 so likely dominate control of the Cu ion coordination compared to protein constraints. This conclusion is consistent with computational studies that argue that the oxidized metal site geometry is inherently preferred due to the electronic structure of Cu(II) coordinated by the particular ligand set[21,22,25,26].

The Cu sites in BCPs demonstrate the interplay between the inherent coordination preferences of a metal ion and the strain imposed by the protein environment. Weak metal-ligand interactions such as Cu(I)-methionine which is vulnerable to protein forces are common in metalloproteins[28,63]. For example, Cu(I)-methionine coordination is involved in the monovalent copper sites in blue and green copper protein families and multivalent CuA sites of cytochrome c oxidase and nitrous oxide reductase that mediate biological electron transport[64]. Methionine ligands are also found in non-enzyme copper binding proteins that serve as chaperones or receptors in metal homeostasis[65–67]. The susceptibility of the weak interactions to even subtle long-distance structural perturbations affords a physical mechanism for allosteric regulation.

We do point out that direct interactions with the secondary sphere are known to be critical for the maintenance of the oxidized site[6,10,16,17], consistent with our inability to generate N40A holoprotein. NMR spectroscopy indicates perturbation by N33A to Asn40 and to both the backbone and sidechain of Cu ligand His39 in the reduced state. The interaction between His39 and the Cu ion thus also likely contributes to the altered Cu(I) coordination and resulting perturbation of redox properties. We have not yet attempted to analyze the histidine ligands by C-D spectroscopy because our experience has found the IR absorptions of deuterated histidine analogs very weak. However, the C-D absorptions may be narrower and easier to detect when the amino acid serves as a metal ligand, like those of $d_2$Cys (Fig. 4a and S8).

Although BCPs are among the best-studied family of metalloproteins and a classic system for illustrating metal site tuning by proteins, the mechanisms underlying the control remain to be fully elucidated. Well-established is the importance of the set of amino acids provided as metal ligands and maintenance of the primary coordination sphere by direct interactions with residues that form a secondary sphere. Here we show how, more globally, the cupredoxin structure also constitutes an outer sphere that contributes to the redox properties. The study provides experimental evidence that the structure of Pc functions to restrict the axial methionine ligand to reduce its interaction with the reduced Cu(I) ion, thereby settling a long-standing question about entatic control of the Cu site in BCPs.

## Methods

**Protein expression and purification.** Plasmid pEAP encoding *Nostoc* Pc was constructed by Miguel A. de la Rosa (Universidad de Sevilla) and was a gift of Marcellus Ubbink (Leiden University). To enable unique labeling at Met97 with (methyl-$d_3$)methionine via expression in defined media, the sequence was modified to include the single methionine residue by the introduction of M66L[37]. The sequence also was modified to encode the precursor MAA-, which is not included in the residue numbering, to promote the removal of Met1 and Ala2 by methionine aminopeptidase. This construct is referred to in this work as wt Pc. Site-directed mutagenesis for the generation of S9A, N33A, N34A, and N40A Pc was performed using QuikChange XL following standard protocols (Agilent, Santa Clara, CA). Mutations were confirmed by sequencing (Quintara Biosciences, Cambridge, MA). The plasmid encoding Pc was cotransformed with plasmid pMetAP encoding methionine aminopeptidase into JM15(DE3) cysteine auxotrophic *E. coli*.

$d_2$Cys89/$d_3$Met97 labeled Pc was expressed in defined media containing M9 salts (3 g/L KH$_2$PO$_4$, 6 g/L K$_2$HPO$_4$, 1 g/L NH$_4$Cl, 0.5 g/L NaCl, 3 mg/L CaCl$_2$), 400 mg/L each amino acid excluding cysteine and methionine, 4 g/L glycerol, 200 μM CuSO$_4$, 50 μg/mL kanamycin, 100 μg/mL ampicillin, 1 mM MgSO$_4$, 1.5 μM thiamine, 140 mg/L $d_4$-DL-cystine, and 50 mg/L (methyl-$d_3$)-methionine (Cambridge Isotopes, Tewksbury, MA). Cells were grown at 37 °C and 250 rpm to OD$_{600}$ of 0.6, and expression was induced by the addition of 1 mM isopropyl β-D-1-thiogalactopyranoside (IPTG). Expression was allowed to continue under the same conditions for 4–5 h, and cells were harvested by centrifugation at 7550 *g* for 25 min.

Cells were resuspended in 2 mM potassium phosphate, pH 7 with 200 μM CuSO$_4$, 1 mM phenylmethylsulfonylfluoride, 20 mM NaCl, 2 mM MgCl$_2$, and 2 U/mL benzonase, and lysed by sonication (Q500, Qsonica, Newtown, CT). The lysate was clarified by centrifugation at 31,000 *g* for 25 min and dialyzed against 2 mM potassium phosphate, pH 7, with 200 μM CuSO$_4$ in an open container until the lysate turned a deep blue color. This typically required 1–2 days but required as long as 4–5 days for the N33A Pc. Following dialysis, the lysate was purified using S-sepharose (BioRad, Hercules, CA), eluting by a gradient of 2–200 mM potassium phosphate. Blue fractions containing oxidized Pc are typically eluted as one resolved peak. Samples were concentrated, exchanged into 2 mM potassium phosphate, pH 7, and used immediately for analysis unless indicated. Samples subject to freeze/thaw treatment were brought to 25% glycerol and flash-frozen in liquid nitrogen and stored at −80 °C. Mass spectra were acquired on a Bruker Autoflex III Smartbeam MALDI-ToF spectrometer in α-cyano-4-hydroxycinnamic acid matrix (Fig. S4).

**IR spectroscopy.** Pc was concentrated to ~2 mM and loaded between two CaF$_2$ windows separated by a Teflon spacer with a thickness of 25.4 μm. Cu(I) Pc was generated by addition of 50 equivalents of dithiothreitol or 200 equivalents of sodium ascorbate, resulting in the loss of the blue color. IR spectra were collected at 2 cm$^{-1}$ resolution on a Cary 670 FT IR (Agilent, Santa Clara, CA) equipped with an MCT detector. A bandpass filter (BBP-4000-5000 nm, Spectrogon, Mountain Lakes, NJ) was introduced between the sample and detector. 4000 (Cu(II)) or 1500 (Cu(I) and apoprotein) double sided interferograms were averaged. Samples of the buffer solution served as a reference. Slow baseline variation was removed by fitting a polynomial function. Each C-D absorption was fit to a Gaussian function. Additional details about IR data processing and analysis are provided in Supplementary Methods. Reported spectra are averages of three replicate samples.

Samples for IR analysis of amide I vibrations were prepared by dilution of the 2 mM protein stock into 2 mM potassium phosphate in D$_2$O, pH 7 to a final concentration of 100 μM. Samples were incubated in the D$_2$O buffer for 2–4 h prior to analysis at room temperature. Spectra were collected as described above but with a long pass filter (4300 nm LWP, Andover, Salem, NH) replacing the bandpass filter. Reported amide spectra are averages of two replicate samples.

**NMR spectroscopy.** $^{15}$N-labeled Pc was expressed in BL21(DE3) *E. coli* by the same method with the following modifications. Cells were grown in minimal media containing 3 g/L KH$_2$PO$_4$, 6 g/L K$_2$HPO$_4$, 1 g/L $^{15}$NH$_4$Cl, 0.5 g/L NaCl, 4 g/L glycerol, 50 μg/mL kanamycin, 100 μg/mL ampicillin, 2 mM MgSO$_4$, 100 μM CaCl$_2$, 3 μM thiamine, and basal medium Eagle vitamins. Cells were grown in 500 mL media in a 2 L flask, and when OD$_{600}$ reached 0.7, 100 μM CuSO$_4$ was added along with 1 mM IPTG to induce expression. NMR samples contained 300 μM Pc with 2 mM potassium phosphate, pH 7, 3 mM sodium ascorbate, and 10% v/v D$_2$O, with 0.3 mM 2,2-dimethyl-2-silapentanesulfonic acid as an internal reference. Details regarding data collection and processing are provided in Supplementary Methods.

**Redox titrations.** $E_m$s was determined by chemical redox titration followed by visible spectroscopy[68]. A spectrum was collected of 5 μM Cu(II) Pc in 20 mM HEPES, 50 mM sodium chloride, pH 6.8, 1.5 mM potassium ferricyanide, 10 μM diphenylamine sulfonic acid, and 2.5 μM ruthenium hexamine. Difference spectra then were collected upon titration with aliquots of 50 mM sodium ascorbate. The absorbance at 600 nm (oxidized Pc) was used to calculate the ratio of reduced and oxidized Pc. During the titration, the potential of the solution was measured using an Ag/AgCl electrode (MTC301, Hach, Loveland, CO) filled with 3 M KCl saturated with AgCl. A linear fit to the Nernst equation provides the $E_m$ (Fig. S6). Additional detail regarding the determination of $E_m$ from these spectra is included in Supplementary Methods.

**Reporting summary**. Further information on research design is available in the Nature Portfolio Reporting Summary linked to this article.

## Data availability

The protein structure used in this study was obtained from the RCSB Protein Data Bank (2GIM). Raw data associated with chemical redox titrations are available as Supplementary Data File 1. NMR spectra are available in Supplementary Data File 2. FTIR data used in this study are available as Supplementary Data File 3. Data supporting this study are also available by request to the corresponding author.

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

## Acknowledgements

This work was supported through the US Department of Energy, Office of Science (DE-FOA-0000751 & DE-SC0022530). The authors are grateful to Miguel A. De la Rosa and Marcellus Ubbink for plasmid pEAP. Protein structural graphics were produced using UCSF Chimera. Chimera is developed by the Resource for Biocomputing, Visualization, and Informatics at the University of California, San Francisco (supported by NIGMS P41-GM103311).

## Author contributions

C.C.M., B.E.L., S.G.E., E.M.M., M.H.M., Y.W., and N.M.G. performed the experiments and C.C.M. analyzed the data. K.A.E. and D.P.G. provided guidance on the collection and analysis of NMR spectra. M.C.T. designed the research, and C.C.M. and M.C.T. wrote the manuscript. All authors have given approval to the final version of the manuscript.

## Competing interests

The authors declare no competing interests.
