## [Peer Review File · Communications Chemistry]

Reviewers' comments:

Reviewer #1 (Remarks to the Author):

The major claim of this paper is that the redox potential of the Cu site in plastocyanin is determined primarily by the interaction between the active site methionine and the reduced Cu(I) state. Overall, the general premise is interesting and creative (using C-D bonds to access allosterically-induced geometric and reduction potential changes to a metal binding site) and sufficient detail is provided within the methods sections. That said, I have some concerns, enumerated in more detail below, about the selection of the specific residues to probe.

My most significant concerns relate to the fact that the paper is predicated quite strongly on a crystal structure (2CJ3) that has not been published in a peer-reviewed journal, and whose PDB entry does not include electron density maps, making it a sub-ideal choice to use to select side chains to mutate. From a quick search of the PDB, it looks like there is at least one structure of the same protein for which structure factors and an associated paper are available (2GIM) – I think this would be a more appropriate model to use. Another very important caveat to keep in mind regarding protein crystal structures is that often the exact orientation of loops are dictated perhaps as much by crystal packing constraints as by hydrogen bonding; structures induced by crystal packing may not adequately mirror experiments in solution. I think it would be helpful to compare how closely the loop regions within different plastocyanin structures superpose on one another, as well as how well the specific H-bonding interactions from 2CJ3 are conserved among other structures.

Somewhat related to the above, it would also be helpful to include a rationale for why the Nostoc homolog was chosen (as well, perhaps, for some discussion about how extrapolatable the findings from this paper would be to other plastocyanin homologs).

OTHER SUGGESTIONS

It might be nice to include some discussion comparing the role of the Met axial ligand in plastocyanin in tuning redox potential with other closely related copper sites, as this could make the paper of broader interest. For example: <https://doi.org/10.1021/ja0731221>; [10.1021/ja0501114](https://doi.org/10.1021/ja0501114))

Figure numbers are not referenced in order within text

Fig S2 – would be helpful for the reader to include the expected masses within the figure legend.

Fig S5 – table doesn't seem to match the spectra (e.g. It looks like there is a higher absorbance for N34A than WT but supposedly more Cu in WT? Clarify how you measured this)

Table 1 – doesn't seem that there is statistical difference between WT and S9A, contrary to what is described in the results.

Reviewer #2 (Remarks to the Author):

The authors describe a relatively novel infrared spectroscopic method for interrogating blue copper proteins using selective isotopic (C/D) substitution of critical methionine and cysteine ligands at the metal site, for both wild-type protein and a range of variants selected to perturb hydrogen bonding between loops of the cupredoxin fold. In agreement with previous reports, they find little if any change in measured properties of the metal site cysteine ligand in the range of variants studied, and report more significant changes in axial methionine ligation. Alongside the spectroscopic data, potentiometric investigation of the reduction potentials of wild-type and variant proteins are presented that suggest the reduction potential of the metal site is shifted to more positive values in the variants, with the greatest effect found for the N33A variant.

The use of C/D exchange to allow observation of specific amino acids through use of the water 'solvent window' is well-discussed elsewhere in the literature. Although the approach taken here is not novel, the application is of sufficient interest to highlight how similar non-natural amino acids could be used in other redox proteins. Therefore the work should be of general interest beyond the immediate community interest in blue copper proteins.

The data presented in the paper have largely been carried out in a convincing manner, although there are a few instances where more details about the FTIR spectra (resolution, for e.g.?) and potentiometric titration (KCl concentration in reference electrode, conversion - or not - of potentials to NHE, etc.) would be welcome. However, it would have been useful to see raw data for the potentiometric titrations as it is not immediately clear that all of the differences in measured reduction potentials are statistically significant. There are also inconsistencies in sample preparation for the reduced proteins for FTIR study that are not explained (identity and quantity of reductant, for e.g.).

I wonder if the authors considered alternative explanations for the increase in FWHM for the reduced variant d3Met197 FTIR? There is good overlap on the low wavenumber side of the band, which could suggest multiple conformations in the variant spectra (i.e. rather than broadening and a peak shift, could this be a second CD3 band centered at higher wavenumber present in varying proportions?). The peak shifts reported are also very small - certainly within the spectral resolution. I agree that it is possible to define a peak center with better accuracy than the spectral resolution, but it is not clear that this is justified in this case without seeing the fitted data. This would also address my earlier concern about the symmetry of the band in the reduced state of the variant spectra. There seem to be similar changes in FWHM in the oxidised protein d3Met197 spectra that are not discussed (although the effect here is smaller).

Although some of the pioneering work of Yi Lu is referenced in the manuscript, a more detailed discussion of their work in the context of the present study would be useful. In particular as the effect of unnatural axial 'methionine' coordination on redox potential has been discussed in detail - J. Am. Chem. Soc. 2006, 128, 15608. This includes discussion of strength of axial ligation, hydrophobicity, protein folding, and hydrogen bonding interactions and so a more in depth comparison to the present data may be warranted (even if the 'methionine' substitutions in the work of Lu and co-workers is somewhat more extreme than that reported here).

Although the authors have reported FTIR spectra in the amide region as proof that the overall structure has not been affected by the mutations, it is not entirely clear that this should be the case. Reporting of second derivative spectra would help here, as this could highlight subtle amide changes in more detail. In any case, if the major changes will be to the looped sections of the proteins it is unclear that FTIR will be the most sensitive technique to assess subtle changes. Have the authors attempted complementary EPR, XAS, crystallographic studies, for example? (Of course, I acknowledge there is no guarantee these will show subtle changes either, but could help to give a more clear picture.)

Overall I think that this work could lead to enhanced understanding of redox proteins through use of the method in other redox systems. However, there are limitations due to the relative insensitivity of C-D bands to subtle changes in coordination (in comparison to other commonly used functional groups in non-natural amino acids such as in cyanophenylalanine for example, where spectral changes can be more obviously diagnostic of local structure).

Reviewer #3 (Remarks to the Author):

In the current manuscript N. M. Garcia, M. C. Thielges and coworkers investigate a blue copper protein, Nostoc plastocyanin (Pc), by means of infrared (IR) spectroscopy in combination with selective labeling with carbon–deuterium (C–D) bonds of the Met (d3Met97) and Cys (d2Cys89) copper ligands. The authors analyzed how perturbations in the H-bonding network of some loops of the cupredoxin fold affect the active site, monitoring changes in the $\text{Cu}^{\text{II}} \rightarrow \text{Cu}^{\text{I}}$ redox potential (E_m) as well as changes in the C–D vibrational spectra of Cys and Met ligands. Three protein variants were recombinantly produced comprising the exchange of Ser9, Asn33 and Asn34 with alanine residues and all showed increased E_m values, suggesting that the reduced state for all constructs is selectively stabilized. Surprisingly, the IR data of d2Cys89 (asymmetric C-D stretching vibrations) for all protein variants were indistinguishable from wild type construct in both oxidized and reduced states, suggesting that the covalent bond between Cu and the S of Cys89 is not perturbed by the amino acid exchanges. On the contrary, the absorptions of the symmetric C-D stretch of d3Met97 appeared affected by the amino acid alterations. Significantly, all protein variants exhibited an increase in the linewidth of the C-D d3Met97 bands in the reduced state and a slight shift to higher energies. The authors, therefore, suggested that the methionine ligand interacts more strongly with the Cu^{I} ion in the protein variants, envisaging that the cupredoxin fold of Pc regulates the redox properties of the active site restricting the Met97 to reduce its interaction with the reduced copper ion.

I have followed previous works from the authors on Pc exploiting Cys/Met C-D labelling for IR spectroscopy (JACS 2016, PCCP 2022). They gained considerable information on the copper active site, especially in states like Cu^{I} that are not accessible by standard spectroscopies (e.g., EPR). The IR experiments in the current investigation are carefully performed, and the interpretations seem sound. I can anticipate that these data will be of interest to BCP's community. However, the significance and novelty appear rather limited in my opinion as similar conclusions about the interaction of the copper and methionine ligand were already reported in their previous JACS (<https://doi.org/10.1021/jacs.6b03916>). Therefore, the novel information that is gained within this

study might have less impact than previous observations. A key point that would leverage the novelty of the current study would be the site-selective deuteration of other ligands (i.e., His) to gain knowledge on the cofactor plasticity in various redox states and how these residues contribute to tuning the reactivity in the biological systems. Although I am impressed by the quality of IR C-D spectra, the current manuscript might not meet the expectations for a paper in Communications Chemistry and – upon revision (see further comments) – is rather suitable for a more specialized journal.

Further comments:

1. The discussion is considerably long compared to the results section. I would suggest the authors to shorten it. Some sentences might also be shifted to the introduction section to increase readability.
2. Figure 2 contains too many plots, and its figure legend is too short lacking details for the understanding to non-specialists of C-D IR spectroscopy.
3. The paper does not address possible conformational changes (in the holo protein) induced by the point mutations. It is clear by the amide I bands that secondary structural elements are retained but this does not strictly indicate that overall protein conformation is not affected. I have not seen any amide II data in the paper/SI for the oxidized and reduced protein variants. Can the authors comment on that? This band can suggest/indicate conformational changes in proteins which might help to rationalize the increased bandwidths of the C-D data for d3Met97 in the reduced state.
4. Did the author have considered to record spectroelectrochemical IR difference spectra? This would allow them to monitor in situ local changes at the Cu site upon electrochemical pulses, targeting e.g., the C-D bands upon $\text{Cu}^{\text{II}} \rightarrow \text{Cu}^{\text{I}}$ reduction.
5. SI needs some improvement. SI figures are not in chronological order (see below) and often there is no information in the MS of the additional text in the SI. For example, in the SI the authors stated “no consistent changes can be discerned in the 1600-1700 cm^{-1} region associated with amide I vibrations (Fig. S8)” Fig. S8 shows only the IR spectra of d2Cys89 wild type upon incubation with urea. Correct figure is S10.
6. Page S5 about the freeze-thaw treatment. The authors stated that “Mass spectrometry shows no change to indicate chemical modification (Fig. S2). However, in Fig. S2 it is not indicated that data are acquired after a freeze-thaw treatment. Do the authors have additional data to support their statement?
7. page 13. The authors stated in the discussion that C-D bands of d2Cys89 indicating partial reduction upon freezing/thawing are reversible and proteins could be re-oxidized using $\text{Fe}^{\text{III}}(\text{CN})_6$. However, no experimental data are shown.

Response to Reviewers.

We thank the reviewers for their time and efforts in assisting with improvements to our manuscript. Specific comments are reproduced below, followed by our response and description of revisions made to address their concerns. While the revisions to the manuscript were extensive, a copy with marked changes to address specific concerns is provided.

Reviewer #1:

The major claim of this paper is that the redox potential of the Cu site in plastocyanin is determined primarily by the interaction between the active site methionine and the reduced Cu(I) state.

We would like to clarify our claim is that the protein structure plays a role in tuning the redox potential by restraining the methionine in the reduced state. We don't intend to argue that this plays a greater role than the identity of the ligand set or the direct interactions between the protein and coordinating ligands in maintaining the oxidized site. Nonetheless, whether and how the greater protein structure contributes to the properties of the metal site is a long standing question that we contend warrants addressing for a complete understanding of metalloprotein design

Overall, the general premise is interesting and creative (using C-D bonds to access allosterically-induced geometric and reduction potential changes to a metal binding site) and sufficient detail is provided within the methods sections.

We appreciate the positive feedback.

That said, I have some concerns, enumerated in more detail below, about the selection of the specific residues to probe.

My most significant concerns relate to the fact that the paper is predicated quite strongly on a crystal structure (2CJ3) that has not been published in a peer-reviewed journal, and whose PDB entry does not include electron density maps, making it a sub-ideal choice to use to select side chains to mutate. From a quick search of the PDB, it looks like there is at least one structure of the same protein for which structure factors and an associated paper are available (2GIM) – I think this would be a more appropriate model to use. Another very important caveat to keep in mind regarding protein crystal structures is that often the exact orientation of loops are dictated perhaps as much by crystal packing constraints as by hydrogen bonding; structures induced by crystal packing may not adequately mirror experiments in solution. I think it would be helpful to compare how closely the loop regions within different plastocyanin structures superpose on one another, as well as how well the specific H-bonding interactions from 2CJ3 are conserved among other structures.

We appreciate reviewer #1 for pointing out the superior structure to include in this work. We have replaced the structure shown in Fig. 1. The targeted residues appear to engage in hydrogen bonding interactions in both crystal structures 2GIM and 2CJ3. However, we agree with the

reviewer that crystal structures provide only one model of the structure under the particular experimental conditions. We have edited the text (Introduction, pp.5, lns 5-10; Discussion, pp.14 ln. 17 – pp. 15 ln. 3) to more accurately convey that the structures guided our mutations, which were intended, rather than known, to disrupt structure.

We now include in the SI an overlay of the *Nostoc* crystal structure, *Nostoc* NMR structure, and spinach NMR structure, showing that the loop structures are generally conserved (Fig. S1). We also include a sequence comparison to illustrate the conservation of N33 and N34 (Fig. S2) and additional references (refs. 27-29, 32-34, 42) that discuss the conservation of sequence and structure among the BCP family in greater detail. While the structures indicate engagement by the residues in hydrogen bonding, the specific interaction partners vary.

To confirm that the mutations disrupt the structure at the northern face of Pc as intended, we collected the $^1\text{H}^{15}\text{N}$ HSQC spectra of ^{15}N -enriched wt and N33A Pc. Descriptions of these experiments and results are included in the manuscript (pp. 10, pp. 22 lns. 8-17) and SI (pp. S4 lns. 8-17).

Somewhat related to the above, it would also be helpful to include a rationale for why the Nostoc homolog was chosen (as well, perhaps, for some discussion about how extrapolatable the findings from this paper would be to other plastocyanin homologs).

Our choice to study the *Nostoc* homolog is motivated by our broader interest in the biophysics of inter-protein electron transfer. *Nostoc* Pc and its redox partner cytochrome *f* were recommended as well-behaved proteins at high concentration (needed for our IR experiments) by the group that has published the majority of the NMR spectroscopy of Pc and its redox partners (Marcellus Ubbink). We thus established constructs for selective C-D labeling in the *Nostoc* background. The primary coordination sphere, N33, and N34 are absolutely conserved among Pc; therefore, we anticipate our results here will apply to other homologs. Residue Ser9 is less conserved; its inclusion in this study contributes a valuable comparison to the other residues. This information is included in the Introduction and Discussion (pp. 5 lns. 5-10, pp. 14, lns. 19-23).

OTHER SUGGESTIONS

It might be nice to include some discussion comparing the role of the Met axial ligand in plastocyanin in tuning redox potential with other closely related copper sites, as this could make the paper of broader interest. For example: <https://doi.org/10.1021/ja0731221>; [10.1021/ja0501114](https://doi.org/10.1021/ja0501114))

We thank reviewer #1 for the suggestions. We have edited the manuscript to incorporate the suggested references (refs. 61, 62) as well as extend the discussion of methionine as a ligand in other copper proteins. We point out that the interplay of metal preferences and constraint imposed by the protein is likely important for other metalloproteins with weak ligation by methionine, such as the CuA site in cytochrome c oxidase and nitrous reductase (pp. 18 ln. 21 – pp. 19 ln 6). We hope added discussion of these related proteins will help to broaden the scope of interest within the metalloprotein field.

Figure numbers are not referenced in order within text

We have reviewed the manuscript carefully to ensure sequential numbering of figures.

Fig S2 – would be helpful for the reader to include the expected masses within the figure legend.

Expected masses are now included in Fig. S4 showing mass spectra.

Fig S5 – table doesn't seem to match the spectra (e.g. It looks like there is a higher absorbance for N34A than WT but supposedly more Cu in WT? Clarify how you measured this)

We thank the reviewer for pointing out the inconsistency between the text and the data from the spectroscopic assay of Cu(II) content of protein preparations. We carefully repeated these experiments. To better convey the measurement procedure, the description in the SI has been edited (pp. S2 ln. 12 – pp. S3 ln. 9, Fig. S5). To help reduce confusion, Fig. S5 now shows only the spectral window containing the region of analysis for the assay. We do reproducibly find that the assay reports only ~0.7 Cu(II) per polypeptide chain for the wt protein, while ~0.9-1.0 Cu(II) per polypeptide chain for the mutated Pc. We suspect the low value obtained for the wt is due to incomplete chemical denaturation, reducing the total Cu(II) released and detected in solution. We have no evidence otherwise that the wt preparation contains significant apoprotein. We point out that the mass spectra (Fig. S4) show no evidence for the presence of apoprotein in any of the protein preparations. The IR spectra of the C-D probes also show no bands consistent with d_3 Met97 apoprotein (Le Sueur, et al. 2016).

Table 1 – doesn't seem that there is statistical difference between WT and S9A, contrary to what is described in the results.

We thank the reviewer for pointing out this omission. We have revised the manuscript to clearly communicate that the difference between the E_m s determined for WT and S9A are within error (pp. 7 lns. 6-7, pp. 14 lns. 19-20). We however still point out the overall correlation shown between E_m and the upshift in C-D frequency among the set of proteins. The increased strength of the axial ligand-Cu(I) interaction indicated by the C-D probes is chemically consistent with the increase in E_m .

Reviewer #2:

The authors describe a relatively novel infrared spectroscopic method for interrogating blue copper proteins using selective isotopic (C/D) substitution of critical methionine and cysteine ligands at the metal site, for both wild-type protein and a range of variants selected to perturb hydrogen bonding between loops of the cupredoxin fold. In agreement with previous reports, they find little if any change in measured properties of the metal site cysteine ligand in the range of variants studied, and report more significant changes in axial methionine ligation. Alongside the spectroscopic data, potentiometric investigation of the reduction potentials of wild-type and variant proteins are presented that suggest the reduction potential of the metal site is shifted to more positive values in the variants, with the greatest effect found for the N33A variant.

The use of C/D exchange to allow observation of specific amino acids through use of the water 'solvent window' is well-discussed elsewhere in the literature. Although the approach taken here is not novel, the application is of sufficient interest to highlight how similar non-natural amino

acids could be used in other redox proteins. Therefore the work should be of general interest beyond the immediate community interest in blue copper proteins.

The data presented in the paper have largely been carried out in a convincing manner, although there are a few instances where more details about the FTIR spectra (resolution, for e.g.?) and potentiometric titration (KCl concentration in reference electrode, conversion - or not - of potentials to NHE, etc.) would be welcome.

The methods and SI have been edited to include substantially more detail about the procedures for acquisition and analysis of FT IR spectra (pp. 21 ln 15 – pp. 22 ln. 7, pp. S4 ln. 19 – pp. S6 ln. 3) and potentiometric titrations (pp. 22 ln. 18 – pp. 23 ln. 4, pp. S3 ln. 11 – pp. S4 ln. 6).

However, it would have been useful to see raw data for the potentiometric titrations as it is not immediately clear that all of the differences in measured reduction potentials are statistically significant.

Overlays of the Nernst plots of the redox fractions and cell potential for triplicate data sets for wt and all mutant Pc has been added to SI (Fig. S6).

There are also inconsistencies in sample preparation for the reduce proteins for FTIR study that are not explained (identity and quantity of reductant, for e.g.).

Reviewer #2 is keen to note that reduced Pc was generated by addition of 200 equivalents sodium ascorbate, or in a couple cases, 50 equivalents of dithiothreitol (DTT). Ascorbate is our preferred reductant due to concerns about displacement of the Cys89 thiol ligand by dithiothreitol. Some spectra of wt and N33A Pc however were collected of reduced proteins prepared with DTT. We point out that no evidence for displacement of the Cys89 ligand was evident in the spectra of samples reduced by dithiothreitol, as the d_2 Cys89 probe absorption did not shift to the frequency characteristic of the unligated residue (Fig. S8). The spectra were identical between Cu(I) Pc samples prepared by both reagents. We thus decided to include data for Cu(I) Pc generated by both reductants in this work and were exact in the description of the sample preparation. Additional detail to explain the alternate reductants is added to the SI (pp. S4 ln. 22 – pp. S5 ln. 6).

I wonder if the authors considered alternative explanations for the increase in FWHM for the reduced variant d3Met197 FTIR? There is good overlap on the low wavenumber side of the band, which could suggest multiple conformations in the variant spectra (i.e. rather than broadening and a peak shift, could this be a second CD3 band centered at higher wavenumber present in varying proportions?). The peak shifts reported are also very small - certainly within the spectral resolution. I agree that it is possible to define a peak center with better accuracy than the spectral resolution, but it is not clear that this is justified in this case without seeing the fitted data. This would also address my earlier concern about the symmetry of the band in the reduced state of the variant spectra.

While we do not think a more complex fit model is justified, we agree that greater attention to this possibility in the manuscript is warranted. We added Fig. S10 showing results from fitting the absorptions to a sum of more than one Gaussian function. When none of the parameters (amplitude, linewidth, frequency) of the two Gaussian functions are restricted, the spectra best fit

to a band with nearly the same frequency as results from the single Gaussian fit and a second band of only 2-6% relative area. When the frequency and linewidth of one band is fixed to the absorption found for wt, the spectra best fit with a second band of substantial area, although the relative areas determined for the two bands have high error (Table S3). In any case, the frequency of this second band increases among mutants similarly as found for the single Gaussian fit, showing the same correlation with E_m .

There seem to be similar changes in FWHM in the oxidised protein d3Met197 spectra that are not discussed (although the effect here is smaller).

We have added Table S2 reporting the FWHM determined for d_3 Met97 of oxidized Pc. As noted, the variance is small. The FWHM do increase as S9A < N34A < N33A, but wt is similar to N34A. The variance in the d_3 Met97 spectra of oxidized Pc is pointed out in the manuscript (pp. 12 lns. 15-16, pp. 15 lns. 20-22).

Although some of the pioneering work of Yi Lu is referenced in the manuscript, a more detailed discussion of their work in the context of the present study would be useful. In particular as the effect of unnatural axial 'methionine' coordination on redox potential has been discussed in detail - J. Am. Chem. Soc. 2006, 128, 15608. This includes discussion of strength of axial ligation, hydrophobicity, protein folding, and hydrogen bonding interactions and so a more in depth comparison to the present data may be warranted (even if the 'methionine' substitutions in the work of Lu and co-workers is somewhat more extreme than that reported here).

We thank reviewer #2 for pointing out the about the multiple ways the properties of Met97 and its coordination of the Cu ion may influence the redox properties. Although we feel an extended discussion of the work would depart from the focus, we have incorporated additional results from Lu's work to enrich the discussion. In particular, the role of increased hydrophobicity on the E_m shown by axial ligand replacement by unnatural amino acids could similarly act as an alternately folded environment surrounding the mutant Pc metal site and contribute to the variation in E_m observed (pp. 17 lns 18-23, pp 18 lns 9-14). Most significantly, our argument that stronger axial ligation underlies the higher E_m s of the mutant Pc at first glance might be construed at odds with Lu's and other's work showing that substitution of Met97 with ligands of higher donor strength leads to lower E_m . The higher E_m s for the mutants studied here of course arises because the oxidized state is unaffected. Thus, our results indicate that the protein constraint only plays a relatively significant role for weak metal-ligand interactions (pp. 18 lns. 16-20). We appreciate the comment of reviewer #2 for helping us to lay out our findings more clearly.

Although the authors have reported FTIR spectra in the amide region as proof that the overall structure has not been affected by the mutations, it is not entirely clear that this should be the case. Reporting of second derivative spectra would help here, as this could highlight subtle amide changes in more detail. In any case, if the major changes will be to the looped sections of the proteins it is unclear that FTIR will be the most sensitive technique to assess subtle changes. Have the authors attempted complementary EPR, XAS, crystallographic studies, for example? (Of course, I acknowledge there is no guarantee these will show subtle changes either, but could help to give a more clear picture.)

Prompted by this suggestion, we have acquired $^1\text{H}^{15}\text{N}$ HSQC spectra of ^{15}N enriched wt and N33A Cu(I) Pc. We felt this approach most effective for assessing small structural changes between loop regions of the reduced proteins. Indeed, overlays of the spectra indicate chemical shift perturbation at residues spanning from N33A to the M97 at the Cu site. We believe these experiments have substantially strengthened our argument that the mutations lead to disruption of the loop structure of the reduced state of Pc.

Additionally, we have included second derivative spectra of the Amide I absorptions of Cu(II) and Cu(I) spectra as suggested by the reviewer (Fig. 2b, c). These spectra similarly indicate minimal perturbation.

Overall I think that this work could lead to enhanced understanding of redox proteins through use of the method in other redox systems. However, there are limitations due to the relative insensitivity of C-D bands to subtle changes in coordination (in comparison to other commonly used functional groups in non-natural amino acids such as in cyanophenylalanine for example, where spectral changes can be more obviously diagnostic of local structure).

We kindly point out that the analyzed C-D mode of d_3 methionine and CN of cyanophenylalanine have similar sensitivity. For example, the probes show almost identical solvatochromatic shifts. The C-D frequency of d_3 methionine shifts 8.1 cm^{-1} from water to toluene (Thielges et. Al., 2008, <https://doi.org/10.1021/ja0779607>), whereas the CN shifts 8.7 cm^{-1} from water to tetrahydrofuran (Getahun et. Al., 2003, <https://doi.org/10.1021/ja0285262>). Most importantly, we stress that C-D bonds are nonperturbative probes. Introduction of a CN group at the metal site would undoubtedly alter the metal site properties so as to provide irrelevant insight.

Reviewer #3:

In the current manuscript N. M. Garcia, M. C. Thielges and coworkers investigate a blue copper protein, Nostoc plastocyanin (Pc), by means of infrared (IR) spectroscopy in combination with selective labeling with carbon–deuterium (C–D) bonds of the Met ($d_3\text{Met}97$) and Cys ($d_2\text{Cys}89$) copper ligands. The authors analyzed how perturbations in the H-bonding network of some loops of the cupredoxin fold affect the active site, monitoring changes in the $\text{CuII} \rightarrow \text{CuI}$ redox potential (E_m) as well as changes in the C–D vibrational spectra of Cys and Met ligands. Three protein variants were recombinantly produced comprising the exchange of Ser9, Asn33 and Asn34 with alanine residues and all showed increased E_m values, suggesting that the reduced state for all constructs is selectively stabilized.

Surprisingly, the IR data of $d_2\text{Cys}89$ (asymmetric C-D stretching vibrations) for all protein variants were indistinguishable from wild type construct in both oxidized and reduced states, suggesting that the covalent bond between Cu and the S of Cys89 is not perturbed by the amino acid exchanges. On the contrary, the absorptions of the symmetric C-D stretch of $d_3\text{Met}97$ appeared affected by the amino acid alterations. Significantly, all protein variants exhibited an increase in the linewidth of the C-D $d_3\text{Met}97$ bands in the reduced state and a slight shift to higher energies. The authors, therefore, suggested that the methionine ligand interacts more strongly with the CuI ion in the protein variants, envisaging that the cupredoxin fold of Pc regulates the redox properties of the active site restricting the Met97 to reduce its interaction with the reduced copper ion.

I have followed previous works from the authors on Pc exploiting Cys/Met C-D labelling for IR spectroscopy (JACS 2016, PCCP 2022). They gained considerable information on the copper active site, especially in states like CuI that are not accessible by standard spectroscopies (e.g., EPR). The IR experiments in the current investigation are carefully performed, and the interpretations seem sound. I can anticipate that these data will be of interest to BCP's community. However, the significance and novelty appear rather limited in my opinion as similar conclusions about the interaction of the copper and methionine ligand were already reported in their previous JACS (<https://doi.org/10.1021/jacs.6b03916>). Therefore, the novel information that is gained within this study might have less impact than previous observations. A key point that would leverage the novelty of the current study would be the site-selective deuteration of other ligands (i.e., His) to gain knowledge on the cofactor plasticity in various redox states and how these residues contribute to tuning the reactivity in the biological systems. Although I am impressed by the quality of IR C-D spectra, the current manuscript might not meet the expectations for a paper in Communications Chemistry and – upon revision (see further comments) – is rather suitable for a more specialized journal.

We respectfully disagree with reviewer #3's view that this current study lacks novelty distinct from our 2016 JACS publication. The objective of that prior work was to establish C-D bonds as probes of the Met97 side chain interaction with the Cu ion of Pc. The publication reported the sensitivity of the probe to redox state and metal substitution, directed at deconvoluting the spectral changes from ionic and covalent changes in the metal-ligand bonding.

The motivation for establishing C-D probes of Met97 and Cys89 is to use them to better understand how proteins tune metal properties - the objective of the current work. BCPs have served as model systems for elaborating the hypothesis that proteins tune metal properties by controlling the coordination geometry. We take advantage of the established C-D probes to access the reduced metal site and provide experimental evidence for resolving a long-standing question about entatic control by the cupredoxin structure of BCPs. Further, the work illustrates how parts of the protein distant from the metal site contribute to the outer coordination sphere and impact the redox properties. Such sensitivity could be leveraged in other proteins for studying long-distance allosteric control (described pp. 18 ln 21 – pp. 19 ln. 6). A comprehensive understanding of BCPs as model proteins will be broadly informative for research into physical mechanisms governing metalloprotein function in general.

Further comments:

1. The discussion is considerably long compared to the results section. I would suggest the authors to shorten it. Some sentences might also be shifted to the introduction section to increase readability.

The manuscript has been extensively revised. As suggested, we have attempted to shift background information to the introduction of the paper and eliminate redundancy.

2. Figure 2 contains too many plots, and its figure legend is too short lacking details for the understanding to non-specialists of C-D IR spectroscopy.

Figure 2 has been divided into multiple figures (2, 4, 5) in the main text, as well as Fig. S11. Figure captions have been revised to facilitate comprehension.

3. The paper does not address possible conformational changes (in the holo protein) induced by the point mutations. It is clear by the amide I bands that secondary structural elements are retained but this does not strictly indicate that overall protein conformation is not affected. I have not seen any amide II data in the paper/SI for the oxidized and reduced protein variants. Can the authors comment on that? This band can suggest/indicate conformational changes in proteins which might help to rationalize the increased bandwidths of the C-D data for d3Met97 in the reduced state.

We have performed $^1\text{H}^{15}\text{N}$ HSQC spectroscopy of ^{15}N enriched wt and N33A Cu(I) Pc. We feel this approach addresses the reviewer's concern in a more rigorous manner than would inclusion of amide II spectra, as it can provide residue-specific detail. Overlay of the $^1\text{H}^{15}\text{N}$ HSQC spectra of wt and N33A indicate that the cupredoxin fold is retained in the presence of N33A, the mutation expected to cause the most perturbation. As anticipated, the spectra indicate that the northern loop structures are perturbed by N33A. The NMR data and a discussion of how it expands our understanding of the impact of the mutation on Pc structure and consequently Cu(I)-Met97 coordination has been added to the manuscript (pp. 10, pp. 17 lns. 5-15, pp. S4 lns. 8-17, Fig. 3, Fig. S7).

4. Did the author have considered to record spectroelectrochemical IR difference spectra? This would allow them to monitor *in situ* local changes at the Cu site upon electrochemical pulses, targeting e.g., the C-D bands upon $\text{CuII} \rightarrow \text{CuI}$ reduction.

While collection of spectroelectrochemical difference spectra can be a valuable method for resolving small changes in the IR spectra resulting from varying the redox state, we do not anticipate that these experiments would provide additional information beyond the spectra individually acquired of oxidized and reduced Pc samples. We note that the IR sample cells consist of two closely spaced windows that are sealed for each use, and we do not have the capability to chemically titrate the sample *in situ*.

5. SI needs some improvement. SI figures are not in chronological order (see below) and often there is no information in the MS of the additional text in the SI. For example, in the SI the authors stated "no consistent changes can be discerned in the 1600-1700 cm^{-1} region associated with amide I vibrations (Fig. S8)" Fig. S8 shows only the IR spectra of d2Cys89 wild type upon incubation with urea. Correct figure is S10.

We regret the manuscript formatting was lacking. Care has been taken to ensure sequential numbering in both the main paper and SI, and the error in figure numbering has been corrected. All figure captions in the main paper and SI have been edited to improve completeness and clarity.

6. Page S5 about the freeze-thaw treatment. The authors stated that "Mass spectrometry shows no change to indicate chemical modification (Fig. S2). However, in Fig. S2 it is not indicated that data are acquired after a freeze-thaw treatment. Do the authors have additional data to support their statement?"

Figure S4b has been added showing overlay of mass spectra of fresh protein and following freeze-thaw treatment. Spectra in Fig. S4a are of previously frozen protein; the mass shifts from wt Pc due to mutations are as expected based on masses of substituted side chains.

7. page 13. The authors stated in the discussion that C-D bands of d2Cys89 indicating partial reduction upon freezing/thawing are reversible and proteins could be re-oxidized using FeIII(CN)6. However, no experimental data are shown.

Figure S14 has been added showing recovery of 600 nm absorbance indicative of oxidized Pc upon addition of chemical oxidant potassium ferricyanide to samples of mutated Pc containing reduced population by IR analysis.

REVIEWERS' COMMENTS:

Reviewer #1 (Remarks to the Author):

I think the manuscript is significantly improved. I enjoyed reading the revised version and have no further suggestions!

Reviewer #2 (Remarks to the Author):

The authors have addressed the majority of the original reviewer comments, and provided reasoned counterpoint to the remainder. The addition of new NMR data in support of structural details is particularly welcome.

Reviewer #3 (Remarks to the Author):

In this revised version M. C. Thielges and coworkers addressed convincingly all the comments/questions of the referees. The paper has been substantially rewritten and simplified following referees' suggestions. Besides a detailed point-to-point response, the authors also performed additional experiments which support their working hypothesis and therefore I do support publication of this paper in Communications Chemistry. I would like to apologize if this 2nd round of revision took longer than expected but the lack of all the changes properly tracked in the revised version delayed the revision process as I had to check two different versions of the paper at the same time.

I have one or two minor concerns about the new NMR data.

- 1) In my opinion the manuscript lacks a comprehensive analysis of the NMR data on the ¹⁵N-labeled wt and N33A Pc variant. The technique needs also to be introduced as non-specialists of HSQC might find this section a bit vague.
- 2) I would suggest moving the paragraph about NMR at page 17 (lines 5-15) of the discussion to the results section so the readers can follow major findings of these new data.
- 3) Why do the authors perform NMR just on the reduced forms of wt and N33A Pc variant? In my opinion data on the as-isolated forms would also be beneficial as they could confirm IR C-D findings, i.e., the mutation does not substantially impact the oxidized state.

COMMSCHEM-23-0112A

Response to Reviewers.

We again thank the reviewers for their time and efforts in assisting with improvements to our manuscript. Specific comments are reproduced below, followed by our response and description of revisions made to address their concerns. A copy with highlighted changes to address specific concerns is provided.

REVIEWERS' COMMENTS:

Reviewer #1 (Remarks to the Author):

I think the manuscript is significantly improved. I enjoyed reading the revised version and have no further suggestions!

Reviewer #2 (Remarks to the Author):

The authors have addressed the majority of the original reviewer comments, and provided reasoned counterpoint to the remainder. The addition of new NMR data in support of structural details is particularly welcome.

Reviewer #3 (Remarks to the Author):

In this revised version M. C. Thielges and coworkers addressed convincingly all the comments/questions of the referees. The paper has been substantially rewritten and simplified following referees' suggestions. Besides a detailed point-to-point response, the authors also performed additional experiments which support their working hypothesis and therefore I do support publication of this paper in Communications Chemistry. I would like to apologize if this 2nd round of revision took longer than expected but the lack of all the changes properly tracked in the revised version delayed the revision process as I had to check two different versions of the paper at the same time.

I have one or two minor concerns about the new NMR data.

1) In my opinion the manuscript lacks a comprehensive analysis of the NMR data on the ¹⁵N-labeled wt and N33A Pc variant. The technique needs also to be introduced as non-specialists of HSQC might find this section a bit vague.

Additional discussion of the analysis has been added to the SI (lines 14-23, p. S4). A copy of the manuscript with the edited sections highlighted has been submitted. In addition, an additional panel has been included to Fig. 3.

2) I would suggest moving the paragraph about NMR at page 17 (lines 5-15) of the discussion to the results section so the readers can follow major findings of these new data.

The Results section was edited to introduce NMR (lines 3-4, pg. 9) and summarize the major findings (lines 10-12, pg. 9).

3) Why do the authors perform NMR just on the reduced forms of wt and N33A Pc variant? In my opinion data on the as-isolated forms would also be beneficial as they could confirm IR C-D findings, i.e., the mutation does not substantially impact the oxidized state.

The oxidized state of plastocyanin contains a paramagnetic Cu(II) center that would quench NMR signals of all residues in the proximity. The resonances of residues of interest surrounding the metal center would not be captured by this experiment.